# Characterization and Risk Assessment of PM_2.5_-Bound Polycyclic Aromatic Hydrocarbons and Their Derivatives Emitted from a Typical Pesticide Factory in China

**DOI:** 10.3390/toxics11070637

**Published:** 2023-07-23

**Authors:** Diwei Wang, Shengmin Wu, Xuesong Gong, Tao Ding, Yali Lei, Jian Sun, Zhenxing Shen

**Affiliations:** 1Department of Environmental Science and Engineering, Xi’an Jiaotong University, Xi’an 710049, China; 2The State Key Laboratory of Environmental Assessment and Pollution Control of Pesticides for Environmental Protection, Nanjing Institute of Environmental Sciences, Ministry of Ecological Environment, Nanjing 210042, China; dingtao@xpu.edu.cn; 3School of Environmental and Chemical Engineering, Xi’an Polytechnic University, Xi’an 710048, China; 4Key Lab of Geographic Information Science of the Ministry of Education, School of Geographic Sciences, East China Normal University, Shanghai 200241, China; yllei@geo.ecnu.edu.cn

**Keywords:** PAHs, pesticide factory, temporal variation, toxicity, health risk

## Abstract

Polycyclic aromatic hydrocarbons (PAHs) and their derivatives have received extensive attention due to their negative effects on the environment and on human health. However, few studies have performed comprehensive assessments of PAHs emitted from pesticide factories. This study assessed the concentration, composition, and health risk of 52 PM_2.5_-bound PAHs during the daytime and nighttime in the vicinity of a typical pesticide factory. The total concentration of 52 PAHs (Σ_52_PAHs) ranged from 53.04 to 663.55 ng/m^3^. No significant differences were observed between daytime and nighttime PAH concentrations. The average concentrations of twenty-two parent PAHs, seven alkylated PAHs, ten oxygenated PAHs, and twelve nitrated PAHs were 112.55 ± 89.69, 18.05 ± 13.76, 66.13 ± 54.79, and 3.90 ± 2.24 ng/m^3^, respectively. A higher proportion of high-molecular-weight (4–5 rings) PAHs than low-molecular-weight (2–3 rings) PAHs was observed. This was likely due to the high-temperature combustion of fuels. Analysis of diagnostic ratios indicated that the PAHs were likely derived from coal combustion and mixed sources. The total carcinogenic equivalent toxicity ranged from 15.93 to 181.27 ng/m^3^. The incremental lifetime cancer risk from inhalation, ingestion, and dermal contact with the PAHs was 2.33 × 10^−3^ for men and 2.53 × 10^−3^ for women, and the loss of life expectancy due to the PAHs was 11,915 min (about 0.023 year) for men and 12,952 min (about 0.025 year) for women. These results suggest that long-term exposure to PM_2.5_ emissions from a pesticide factory has significant adverse effects on health. The study results support implementing the characterization of PAH emissions from pesticide factories and provides a scientific basis for optimizing the living environment around pesticide factories.

## 1. Introduction

Atmospheric particulate matter of a diameter ≤ 2.5 µm (PM_2.5_) has become a global concern due to its adverse effects on the environment and human health [1,2]. Several studies have shown that areas with high concentrations of PM_2.5_ have significantly higher incidence rates of cancer and of respiratory, cardiopulmonary, and nervous system diseases [3,4]. Polycyclic aromatic hydrocarbons (PAHs) are the main toxic component of atmospheric PM_2.5_. PAHs have attracted widespread attention due to their carcinogenic and mutagenic effects [5,6,7].

PAHs are a group of fused-ring aromatic compounds and can be divided into several subgroups: parent-PAHs (p-PAHs), alkylated-PAHs (a-PAHs), oxygenated-PAHs (o-PAHs), and nitrated-PAHs (n-PAHs) [8]. Most of the related studies on PAHs focus on the 16 priority-controlled PAHs identified by the United States Environmental Protection Agency [9,10,11]. However, several studies have shown that failing to evaluate the toxicity of PAHs other than the 16 priority-controlled PAHs may lead to substantial underestimation of the actual toxicity in complex environmental samples. Samburova et al. [12] found that the 16 priority-controlled PAHs were responsible for only 14.4%, on average, of the benzo(a)pyrene equivalent (BaP_eq_) toxicity in 13 projects. o-PAHs and n-PAHs may be even more toxic than p-PAHs in certain circumstances [13,14,15]. For instance, p-PAHs are known to have indirect mutagenicity [16], whereas several n-PAHs, including 1-nitropyrenen and 2-nitropyrene, have direct mutagenicity [17]. Furthermore, benzo[a]pyrene-6,12-quinone, which is an o-PAH, contributes to the overproduction of intracellular reactive oxygen species to a greater degree than its parent precursor benzo[a]pyrene [18].

PAHs, which are mainly the product of incomplete combustion, are derived from the combustion of various fuels, such as coal, biomass, diesel oil, and gasoline [7,8,19]. PAH derivatives (a-PAHs, o-PAHs, and n-PAHs) are primarily formed through secondary reactions with OH and NO_3_ radicals [20]. Large amounts of PAHs are emitted from industrial sources due to fuel combustion, solvent use, and production processes. These emissions have serious effects on the local environment and on human health [21,22]. The pesticide factories contribute significantly to air pollutant emissions in the organic chemical industry [23], which is due to the fact that the various volatile and highly toxic substances, such as aromatic hydrocarbons, ketones, and aldehydes, are involved in the production of pesticides [24,25]. Some PAHs and coal-tar-containing PAHs are also important raw materials for pesticide production, which may lead to the emission of PAHs in the production process of enterprises [26]. For example, Jungers et al. [27] investigated the levels of the 16 priority-controlled PAHs of 24 pesticides, and found that the concentration of 11 PAHs exceeded the standards by 50 times, and three of them were greater than the standards by over 1000 times, which resulted in the ratio of the PAHs to the threshold of toxicity being from 2.16 to 8288 times. However, few studies have investigated the characteristics and health hazards of PAH emissions from pesticide factories. According to data from the National Bureau of Statistics of China (http://www.stats.gov.cn/. Last access date: 15 July 2023), the total production of pesticides was 2.148 million tons in China in 2020. The number of pesticide manufacturing factories in China is approximately 1500~1700 [23]. These results indicate that PAH emissions from pesticide factories in China may be severe, and it is important to investigate the concentration and distribution characteristics of PAHs in typical pesticide factories. The present study conducted a sampling campaign to measure daytime and nighttime PM_2.5_ concentrations in the vicinity of a typical pesticide factory in Jiangsu Province, China. A total of 52 PM_2.5_-bound PAH species were quantified. The main objectives of this study are as follows: (1) to determine the concentration, temporal variation, and composition of PM_2.5_-bound PAHs in the vicinity of a pesticide factory; (2) to obtain the distribution characteristics of PAHs and their derivatives; and (3) to evaluate the health risks of exposure to PAHs. This study investigated exposure to PAHs in terms of atmospheric chemistry and human health. The study results may provide a scientific basis for optimizing the living environment around pesticide factories.

## 2. Materials and Methods

### 2.1. Study Site and Sample Collection

Atmospheric PM_2.5_ samples were obtained in the vicinity of a pesticide factory (34.37° N, 118.35° E) in Xuzhou, Jiangsu Province, China. The pesticide factory mainly produces insecticides, fungicides, herbicides, and chemical intermediates. The map near the sampling site is shown in Appendix A. To the west of the pesticide factory is Xindai River. No buildings are present to the northeast of the factory. Approximately 50 m to the southwest of the factory is a wooden board compression factory, and approximately 5 m to the east of the factory is a stone factory. The area has no other chemical factories. The sampling site was downwind of the factory, 10 m away from the boundary of the factory, and approximately 150 m away from the chimney of the factory. The nearest road was 30 m away and had low vehicular traffic (<100 vehicles per day). The weather conditions during the sampling period are shown in Appendix A. During the sampling period, the weather was mainly sunny and the wind direction was mainly easterly. The sampling site was located at the east direction of the factory chimney. Therefore, the sampling site was downwind of the factory. Due to the impact of atmospheric transport, most of the PAHs in the collected samples were mainly coming from the emission of the pesticide factory. It is worth noting that there are a total of 10 villages within 7 km of the pesticide factory, of which the nearest is 1.5 km and the farthest is about 6.4 km away, and the total population of these villages is about 18,000.

At the sampling site near the pesticide factory, a medium-volume PM_2.5_ sampler with a flow rate of 50 L/min was used to collect PM_2.5_ samples on 47 mm prebaked quartz-fiber filters (Whatman, Maidstone, Kent, UK). Samples were collected continuously daily during the daytime (8 a.m. to 8 p.m.) and nighttime (8 p.m. to 8 a.m.) between 15 March and 23 March 2023. Finally, a total of nine daytime samples and eight nighttime samples was collected for further analysis. A field blank sample was collected at the same sampling site. All sampled filters were transported at −18 °C, and stored in a freezer at the same temperature in the laboratory. The quartz-fiber filters were used in accordance with the quality assurance and control protocols described in detail by Cao et al. [28] and Shen et al. [29].

### 2.2. PAH Analysis

A total of 52 PAHs were detected, specifically twenty-two p-PAHs, seven a-PAHs, ten o-PAHs, and twelve n-PAHs (Appendix A). The process by which the PAHs were extracted and quantified is described in several other studies [1,8,19,30]. Briefly, a quarter of each filter sample was extracted in an ultrasonic bath containing 5 mL of dichloromethane and methanol (3:1, *v*/*v*) for 15 min. Water in each extract was removed by adding anhydrous sodium sulfate (Sigma-Aldrich, St. Louis, MO, USA). The solution was concentrated to less than 0.5 mL and dried under a gentle pure nitrogen flow. The particles and filter fiber residues of the concentrated extracts were removed using a syringe filter (0.2 mm pore size, 13 mm diameter; MFS, Dublin, CA, USA). The final solution was then mixed with 25 μL of injection internal standard (fluoranthene-d10, 20 ng/μL) to obtain a total volume of 1 mL. The concentration of PAHs in each filter extract was quantified using a gas chromatography/mass spectrometry system (GC7890/5975MS, Agilent Technology, Santa Clara, CA, USA). Quality assurance and control protocols regarding the PAH analysis have been described by Ho et al. [30] and Zhang et al. [8].

### 2.3. Health Risk Assessment

The potential health effects of exposure to PM_2.5_-bound PAH emissions were assessed using carcinogenic-equivalent toxicity (TEQ), mutagenic-equivalent toxicity (MEQ), incremental lifetime cancer risk (ILCR), and loss of life expectancy (LLE). TEQ and MEQ represent carcinogenic risk at an equivalent concentration of benzo[a]pyrene (BaP) and are widely used to assess the total toxicity of PAHs [6,19,31]. ILCR is a model developed by the United States Environmental Protection Agency that is widely used to quantitatively estimate the risk of cancer caused by exposure to PM_2.5_-bound PAHs, and the model includes inhalation (ILCR_Inhalation_), ingestion (ILCR_Ingestion_), and dermal contact (ILCR_Dermal_) [32,33]. The parameters used in the ILCR model in present study can be found in previous literature [1,32,34,35,36] and are listed in Appendix A. LLE refers to the expected loss of life due to exposure to carcinogenic environmental pollutants [31,37]. Further information and details regarding the calculation methods used in the present study are provided in Appendix A.

## 3. Results and Discussion

### 3.1. Characterization of PM_2.5_-Bound PAHs

The total concentration of 52 PAHs (Σ_52_PAHs) during the study period was 53.04–663.55 ng/m^3^, and the mean concentration was 200.63 ± 159.42 ng/m^3^ (Figure 1). The concentrations of Σ_52_PAHs in the present study are comparable to those reported in two studies examining PAHs from other sources, including motor vehicle emissions at a road tunnel exit (Σ_30_PAHs, 183.40 ng/m^3^) [7] and coal combustion and biomass burning (Σ_21_PAHs, 104 ng/m^3^) [38]. However, the concentrations are higher than those reported in studies assessing ambient air samples in Beijing (Σ_61_PAHs_,_ 59.96 ng/m^3^) [39], Xi’an (Σ_52_PAHs, 65.40 ng/m^3^) [1], and Hong Kong (Σ_46_ PAHs, 4.67 ng/m^3^ [40]. These comparisons suggest that the emissions from the pesticide factory contain an excessive concentration of PAHs. In the present study, twenty-two p-PAHs, seven a-PAHs, ten o-PAHs, and twelve n-PAHs were quantified. The average concentrations were 112.55 ± 89.69 ng/m^3^ (range: 28.11–355.54 ng/m^3^), 18.05 ± 13.76 ng/m^3^ (5.33–58.70 ng/m^3^), 66.13 ± 54.79 ng/m^3^ (17.77–238.73 ng/m^3^), and 3.90 ± 2.24 ng/m^3^ (1.13–10.58 ng/m^3^) for the sum of the concentrations of p-PAHs (Σ_22_PPAHs), a-PAHs (Σ_7_APAHs), o-PAHs (Σ_10_OPAHs), and n-PAHs (Σ_12_NPAHs), respectively.

Although Σ_52_PAHs varied substantially during the sampling period, no significant differences were observed between daytime and nighttime concentrations, which may be related to the production process of the pesticide factory. The basic factory information collected included the environment impact assessment statement, and engineering data, which show that the production system of the factory is continuous operation. In the daytime, Σ_52_PAHs ranged from 53.04 to 663.55 ng/m^3^, with an average concentration of 203.96 ng/m^3^, and during the nighttime, the concentration ranged from 59.63 to 393.41 ng/m^3^, with an average concentration of 196.90 ng/m^3^. The composition proportion of PAHs was stable throughout the sampling period (Figure 2). p-PAHs were the most abundant, accounting for 55.6% of Σ_52_PAHs during the daytime and 56.9% during the nighttime. a-PAHs accounted for 9.5% of Σ_52_PAHs during the daytime and 8.4% during the nighttime. o-PAHs and n-PAHs contributed 33.0% and 1.9%, respectively, to Σ_52_PAHs, with similar contributions during both the daytime and nighttime. The proportion of n-PAHs in this study is higher than that in studies examining PAHs from other sources. For example, n-PAHs accounted for only 0.1% of the total PAHs from biomass burning and coal combustion [8]. The higher proportion of n-PAH emissions may be related to the production process of the pesticide factory.

### 3.2. Distribution of p-PAHs and Their Derivatives

The composition profiles of the PAHs during the sampling period are shown in Figure 3. The PAH species detected in the present study are listed in Appendix A. Among the 22 p-PAH species, BghiP was the most abundant (9.66 ng/m^3^), accounting for 8.6% of the total p-PAH species. Among a-PAH species, 2,6DM-NAP (4.13 ng/m^3^), 9M-ANT (3.31 ng/m^3^), and RET (3.04 ng/m^3^) were the most abundant, all together accounting for 58.1% of the total a-PAH species. Moreover, 9-FO, 9,10-ATQ, and BaAQ were the most abundant o-PAH species, accounting for 17.8%, 16.2%, and 14.2% of Σ_10_OPAHs, respectively. Among n-PAH species, 9N-ANT (24.6%) and 2N-BIP (11.5%) were the most abundant. No obvious differences in the average concentrations of individual PAH species were observed between daytime and nighttime samples (Appendix A).

The PAHs detected in the present study were classified into five groups according to the number of aromatic rings in their molecular structure, which ranged from two to six. The number of rings in each PAH is shown in Appendix A. The distribution of PAHs according to the number of rings is presented in Figure 4. The daily mean mass concentrations of PAHs were as follows: 77.14 ng/m^3^ for 4-ring PAHs, 41.65 ng/m^3^ for 5-ring PAHs, 34.80 ng/m^3^ for 2-ring PAHs, 27.91 ng/m^3^ for 3-ring PAHs, and 19.14 mg/m^3^ for 6-ring PAHs. PAH species were further classified into low-molecular-weight PAHs (i.e., 2-ring and 3-ring PAHs), medium-molecular-weight PAHs (i.e., 4-ring PAHs), and high-molecular-weight PAHs (i.e., 5-ring and 6-ring PAHs) [19]. On average, the medium-molecular-weight PAHs dominated the total PAHs by mass contribution (38.5%), followed by the low-molecular-weight PAHs (31.2%) and the high-molecular-weight PAHs (30.3%). These trend results are similar to those of other studies [7,19,41]. Low-molecular-weight PAHs with high Henry’s constants and vapor pressures mainly exist in the gas phase and are difficult to capture and collect, whereas middle-to-high-molecular-weight PAHs with low Henry’s constants and vapor pressures mainly exist in the particle phase [31,42]. Additionally, high-temperature processes (i.e., the combustion of fuels in industrial boilers in pesticide factories) emit high-molecular-weight PAHs to a greater extent than they emit low-molecular-weight PAHs [43,44].

### 3.3. Diagnostic Ratios of PAHs

Diagnostic ratios of PAHs are applied to identify and distinguish their emission sources [7]. Calculated ratios of the PAHs in the present study are listed in Table 1. From the ratios of ANT/(ANT + PHE) and BaA/(BaA + CHR), it can be inferred that the study area is more likely to be affected by pyrogenic and mixed combustion sources than by other sources [1]. The BaP/BghiP ratio was 0.96 ± 0.25 during the daytime and 0.94 ± 0.38 during the nighttime, corresponding to the characteristic values for emissions from coal combustion [6]. This observation indicates that the pesticide factory possibly uses coal as fuel. Furthermore, the FLA/(FLA + PYR) ratio was close to 0.5 during the sampling period, further indicating the contribution of coal combustion to the emission of the PM_2.5_-bound PAHs. The ratios of IcdP/(IcdP + BghiP), BaP/(BaP + CHR), BbF/BkF, and PYR/BaP corresponded to those for petroleum combustion and diesel/gasoline emissions, suggesting that vehicle emissions contributed to the PAHs in the study area [7,41]. The FLO/(FLO + PYR) ratio in the present study had a similar value (0.29 ± 0.05) during both daytime and nighttime and was less than the value indicating petroleum combustion (0.40–0.50) [1]. Therefore, the source of these PAHs is more likely to be the production processes at the pesticide factory than the vehicle emissions.

Diagnostic ratios of PAHs in the vicinity of the pesticide factory correspond to values that are characteristic of various combustion sources and mixed sources, including coal combustion, petroleum combustion, and diesel emissions. Considering the surrounding environmental conditions in the sampling area, the identified sources are most likely the production processes at the pesticide factory given its use of coal as boiler fuel and the various emissions during its production processes.

### 3.4. Risk Assessment of PAHs from the Pesticide Factory

Toxicity equivalent factor and mutagenic potency factor values for each of the PAHs in the present study were obtained from the literature (Appendix A) [13,15,45,46]. There are limited numbers of PAH species that have TEF and MEF values; the results obtained were only a part of the actual total toxicology. As presented in Figure 5, the ΣTEQ of the PM_2.5_-bound PAHs emitted by the pesticide factory ranged from 15.93 to 181.27 ng/m^3^, with an average value of 61.93 ± 46.46 ng/m^3^. This value exceeds that recommended by the World Health Organization (1 ng/m^3^) [47]. Furthermore, the ΣTEQ of the PAHs in this study is much higher than those of PAHs from ambient air samples in Harbin (9.19 ng/m^3^) [6], Beijing (10 ng/m^3^) [39], Guangzhou (11 ng/m^3^) [48], Xi’an (17.16 ng/m^3^) [1], and Wuhan (7.12 ng/m^3^) [1]. Notably, the proportion of n-PAHs to total PAHs was 1.9%; yet, n-PAHs accounted for 2.6% of the ΣTEQ, indicating that n-PAHs are more toxic than other PAH derivatives [6]. The ΣMEQ ranged from 3.60 to 41.09 ng/m^3^, with a mean value of 14.64 ± 12.41 ng/m^3^. These results are comparable to those observed for ambient air samples in Harbin (12.1 ng/m^3^) [6] and Beijing (13.5 ng/m^3^) [49] but higher than those observed in Nantong (0.09–0.37 ng/m^3^) [31]. Additionally, no significant differences in both ΣTEQ and ΣMEQ values were observed between daytime and nighttime samples (Figure 6). The average ΣTEQ values were 62.38 ± 53.45 ng/m^3^ and 61.41 ± 40.84 ng/m^3^ during the daytime and nighttime, respectively. The average ΣMEQ values were 14.64 ± 12.41 ng/m^3^ and 14.63 ± 9.67 ng/m^3^ during the daytime and nighttime, respectively. The TEQ and MEQ values of individual PAH species are listed in Appendix A. Among the PAHs detected, DBahA was the main contributor to the ΣTEQ, accounting for 38.0% of the ΣTEQ. CPcdP, DBaeP, and BaP were the next main contributors, accounting for 23.1%, 16.5%, and 12.2%, respectively, of the ΣTEQ. The contribution of BaP to the ΣMEQ was dominant (51.7%).

The ILCRs resulting from inhalation, ingestion, or dermal contact with the PAHs in the present study were 2.33 × 10^−3^ for men and 2.53 × 10^−3^ for women (Table 2). According to the United States Environmental Protection Agency, ILCR values between 10^−6^ and 10^−4^ correspond to a low risk of cancer, and ILCR values greater than 10^−4^ correspond to a high risk of cancer [50,51]. The ILCRs in the present study exceed recommended standards (10^−6^). Dermal contact (1.49 × 10^−3^ for men and 1.62 × 10^−3^ for women) and ingestion (8.38 × 10^−4^ for men and 9.11 × 10^−4^ for women) were the main exposure pathways. These pathways were associated with an ILCR 4–5 orders of magnitude higher than the ILCR due to inhalation. A comparison of ILCR caused by p-PAHs and their derivatives revealed that the p-PAHs were the largest contributor (93.6%) to a low risk of cancer (ILCR = 10^−6^ to 10^−4^), followed by o-PAHs and n-PAHs. a-PAHs posed a very low risk of cancer (ILCR ≤ 10^−6^). These results may be due to the high proportion of p-PAHs in the samples and the fact that data for PAHs other than p-PAHs were unavailable [6]. The LLE due to PAHs was 11,915 min (about 0.023 year) for men and 12,952 min (about 0.025 year) for women (Table 3). Among the PAH types, p-PAHs provided the greatest contribution to the LLE (11,169 min for men and 12,141 for women). The LLE due to p-PAHS was 28 times and 33 times higher than the LLE due to o-PAHs and n-PAHs, respectively. The LLE attributed to a-PAHs was 0.51–0.56 min. These results indicate that long-term exposure to PM_2.5_ emissions from pesticide factories results in significant adverse effects on health.

## 4. Conclusions

In the present study, daytime and nighttime PM_2.5_-bound PAHs emitted from a typical pesticide factory were detected. Their concentration, temporal variation, and health risk were assessed. The average concentration of 52 PAHs was 200.63 ± 159.42 ng/m^3^. Although Σ_52_PAHs varied during the sampling period, no significant differences were observed between daytime and nighttime samples. p-PAHs were the most abundant PAHs, accounting for 55.6% of the total PAH concentration. The proportion of high-molecular-weight (4–5-ring) PAHs was higher than that of low-molecular-weight PAHs. The analysis of diagnostic ratios suggested that the emissions from the pesticide factory are caused by production processes and coal combustion. The average ΣTEQ of the PAHs was 61.93 ± 46.46 ng/m^3^. These values are much higher than those observed in ambient air samples, indicating the potential risks of exposure to emissions from the pesticide factory. The ILCR of the PAHs was 2.33 × 10^−3^ for men and 2.53 × 10^−3^ for women. The LLE caused by the ΣPAHs was 11,915 min (about 0.023 year) for men and 12,952 min (about 0.025 year) for women. Among the four types of PAHs, the LLE caused by p-PAHs was the highest (11,169 min for men and 12,141 for women) and was 28 times and 33 times higher than the LLE caused by o-PAHs and n-PAHs, respectively. The LLE caused by a-PAHs was the lowest. The results of this study indicate the presence of elevated concentrations of PAHs in the emissions from the pesticide factory, which could potentially impact the health of individuals working in proximity to these emissions over the long term.

## Figures and Tables

**Figure 1 toxics-11-00637-f001:**
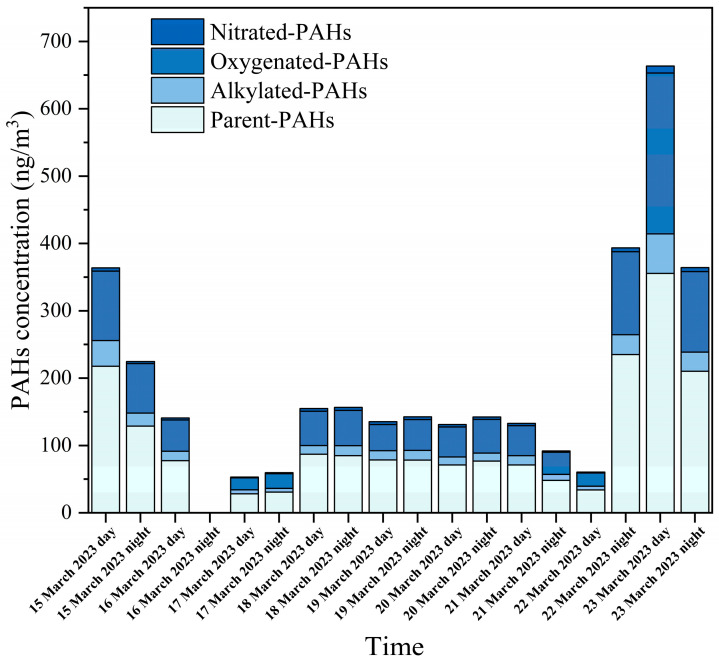
Temporal variation of concentration of PM_2.5_-bound PAHs.

**Figure 2 toxics-11-00637-f002:**
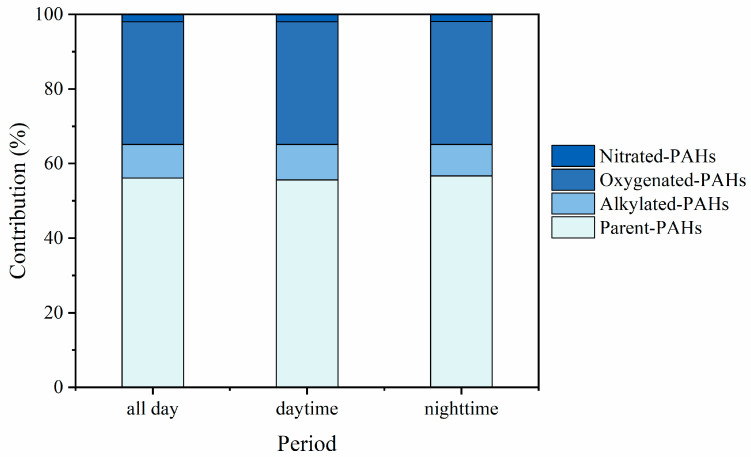
The contribution of parent PAHs and their derivatives during daytime and nighttime.

**Figure 3 toxics-11-00637-f003:**
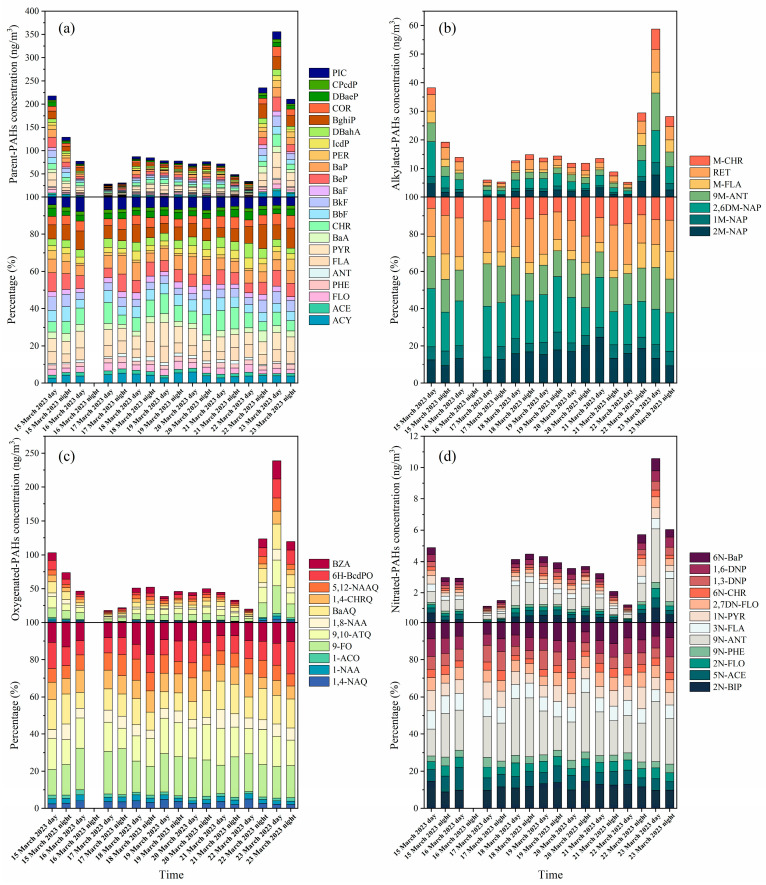
Temporal variation of concentration and composition of PAHs: (**a**) parent-PAHs; (**b**) alkylated-PAHs; (**c**) oxygenated-PAHs; (**d**) nitrated-PAHs.

**Figure 4 toxics-11-00637-f004:**
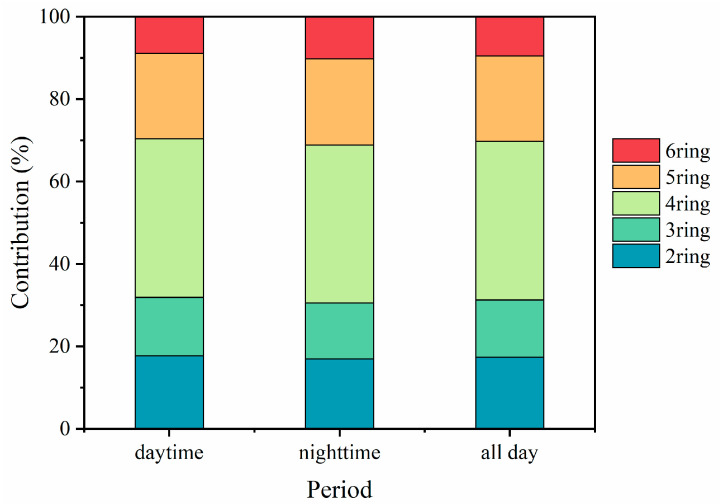
The average contribution of PAHs with different ring numbers during daytime and nighttime.

**Figure 5 toxics-11-00637-f005:**
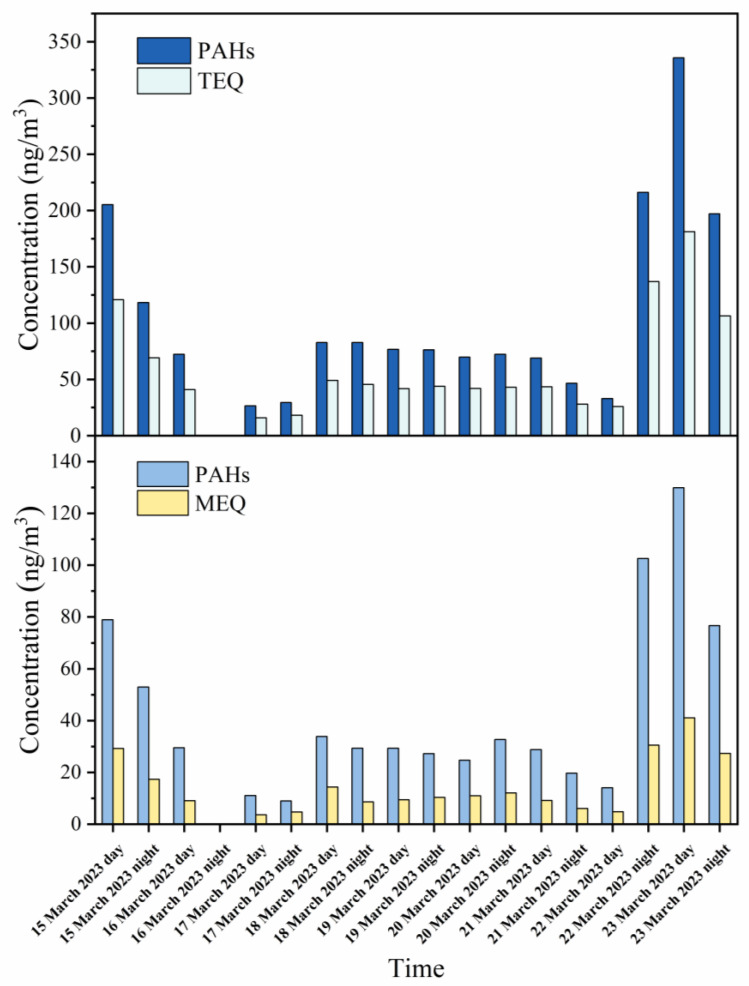
Temporal variation of TEQ and MEQ of PM_2.5_-bound PAHs during the sampling period.

**Figure 6 toxics-11-00637-f006:**
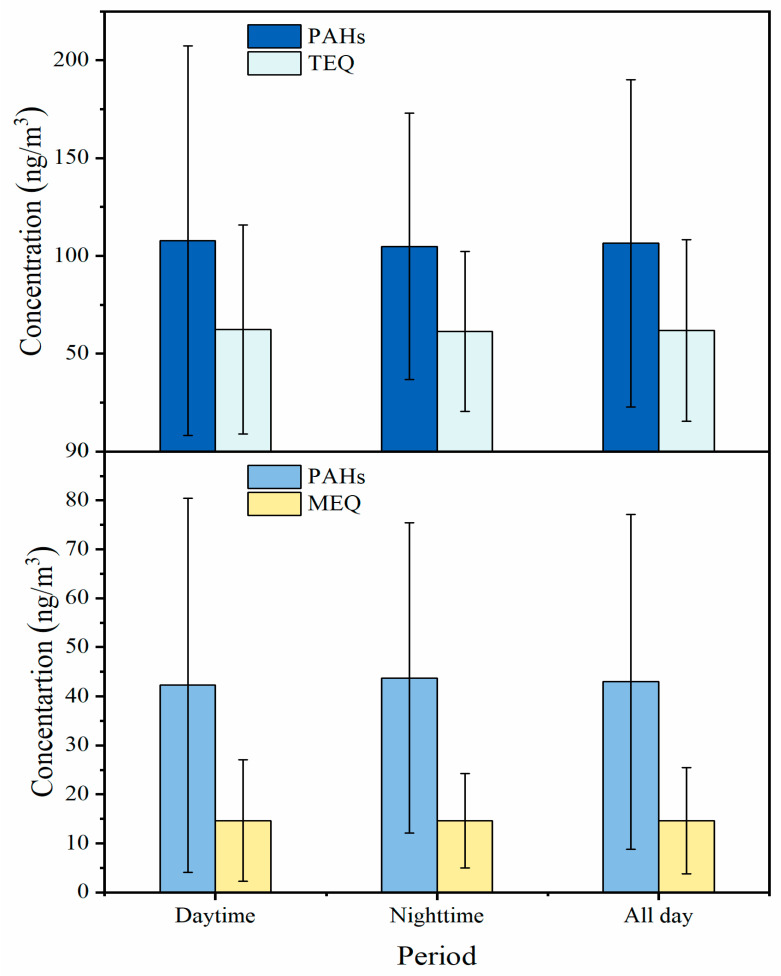
The average TEQ and MEQ values of PM_2.5_-bound PAHs during daytime and nighttime.

**Table 1 toxics-11-00637-t001:** PAHs diagnostics ratios in this study.

Diagnostic Ratios	Daytime	Nighttime	Sources in the Reference
ANT/(ANT + PHE)	0.37 ± 0.08	0.35 ± 0.11	Pyrogenic sources: >0.1 [1]
BaA/(BaA + CHR)	0.29 ± 0.05	0.32 ± 0.11	Mixed source: 0.20–0.35 [1]
FLO/(FLO + PYR)	0.29 ± 0.05	0.29 ± 0.05	Petroleum combustion: 0.40–0.50 [1]
IcdP/(IcdP + BghiP)	0.32 ± 0.06	0.29 ± 0.07	Petroleum combustion: 0.20–0.50 [41]
FLA/(FLA + PYR)	0.42 ± 0.05	0.45 ± 0.04	Coal combustion: 0.40–0.50 [6]
BaP/(BaP + CHR)	0.46 ± 0.09	0.48 ± 0.11	Diesel emission: 0.50 [7]
BbF/BkF	1.27 ± 0.34	1.22 ± 0.37	Diesel emission: >0.5 [7]
BaP/BghiP	0.96 ± 0.25	0.94 ± 0.38	Coal combustion: >0.9 [6]
PYR/BaP	1.24 ± 0.39	1.15 ± 0.46	Gasoline: ~1 [7]

**Table 2 toxics-11-00637-t002:** Incremental lifetime cancer risk (ILCR) caused by exposure to pesticide-factory-emitted PM_2.5_-bound PAHs and their derivatives via inhalation, ingestion, and dermal contact.

		p-PAHs	a-PAHs	o-PAHs	n-PAHs	ΣPAHs
Male adult	Inhalation	5.70 × 10^−8^	2.62 × 10^−12^	2.05 × 10^−9^	1.75 × 10^−9^	6.08 × 10^−8^
	Ingestion	7.86 × 10^−4^	3.62 × 10^−8^	2.83 × 10^−5^	2.42 × 10^−5^	8.38 × 10^−4^
	Dermal	1.40 × 10^−3^	6.43 × 10^−8^	5.03 × 10^−5^	4.29 × 10^−5^	1.49 × 10^−3^
	ILCR	2.18 × 10^−3^	1.00 × 10^−7^	7.86 × 10^−5^	6.71 × 10^−5^	2.33 × 10^−3^
Female adult	Inhalation	5.00 × 10^−8^	2.30 × 10^−12^	1.80 × 10^−9^	1.54 × 10^−9^	5.34 × 10^−8^
	Ingestion	8.54 × 10^−4^	3.93 × 10^−8^	3.08 × 10^−5^	2.63 × 10^−5^	9.11 × 10^−4^
	Dermal	1.52 × 10^−3^	6.99 × 10^−8^	5.47 × 10^−5^	4.67 × 10^−5^	1.62 × 10^−3^
	ILCR	2.37 × 10^−3^	1.09 × 10^−7^	8.55 × 10^−5^	7.29 × 10^−5^	2.53 × 10^−3^

**Table 3 toxics-11-00637-t003:** Loss of life expectancy (LLE, min) caused by exposure to pesticide-factory-emitted PM_2.5_-bound PAHs and their derivatives.

	p-PAHs	a-PAHs	o-PAHs	n-PAHs	ΣPAHs
Male adult	11,169	0.51	403	343	11,915
Female adult	12,141	0.56	438	373	12,952

## Data Availability

The data presented in this study are available on request from the corresponding author.

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
