# Peer review of "Characterization and Risk Assessment of PM_2.5_-Bound Polycyclic Aromatic Hydrocarbons and Their Derivatives Emitted from a Typical Pesticide Factory in China"

_toxics, 2023, doi:10.3390/toxics11070637_

Round 1

Reviewer 1 Report

General Comments onCharacterization and risk assessment of PM2.5-bound polycyclic aromatic hydrocarbons and their derivatives emitted from a typical pesticide factory in China

This study assessed the concentration, composition, and health risk of 52 PM2.5-bound PAHs during the daytime and nighttime in the vicinity of a typical pesticide factory in China. The manuscript was well structured, with good-quality of English writing and figures and tables. The limitation is that the study period is only 8 days, from March 15 to 23, 2023. I suggest “accept after minor revision”.

Specific comments:

1. Line 29: the loss of life expectancy due to the PAHs was 11915 min for men and 12952 min for women. It was suggested to convert the loss of life expectancy of minutes into years.

2. Line 68-69: Few studies have investigated the characteristics and health hazards of PAH emissions from pesticide factories. It was suggested to summarize the importance of examining PAH emissions from pesticide factories in China.

3. Section 2.1. Study site and sample collection. How about the population density around the factoryIt was suggested to briefly introduce the distribution of residents around the factory.

4. Line 147-148: “The concentrations of PAHs were stable throughout the sampling period (Fig. 2).” It’s not “concentration”. It may be the “composition proportion of PAH”.

Author Response

We appreciate the editor and anonymous reviewer for their constructive comments and suggestions, which assist to improve the quality of our manuscript.

Reviewer #1:

General Comments on “Characterization and risk assessment of PM2.5-bound polycyclic aromatic hydrocarbons and their derivatives emitted from a typical pesticide factory in China”.

This study assessed the concentration, composition, and health risk of 52 PM2.5-bound PAHs during the daytime and nighttime in the vicinity of a typical pesticide factory in China. The manuscript was well structured, with good-quality of English writing and figures and tables. The limitation is that the study period is only 8 days, from March 15 to 23, 2023. I suggest “accept after minor revision”.

Specific comments:

  1. Line 29: the loss of life expectancy due to the PAHs was 11915 min for men and 12952 min for women. It was suggested to convert the loss of life expectancy of minutes into years.

Response: Suggestion taken. The sentence has been revised as follows:

P1 Line 27-30:

“The incremental lifetime cancer risk from inhalation, ingestion, and dermal contact with the PAHs was 2.33 × 10−3 for men and 2.53 × 10−3 for women, and the loss of life expectancy due to the PAHs was 11915 min (about 0.023 year) for men and 12952 min (about 0.025 year) for women.”

  1. Line 68-69: Few studies have investigated the characteristics and health hazards of PAH emissions from pesticide factories. It was suggested to summarize the importance of examining PAH emissions from pesticide factories in China.

Response: Suggestion taken. The sentence has been revised as follows:

P2 Line 72-78:

“Some PAHs and coal tar containing PAHs are also important raw materials for pesticide production, which may lead to the emission of PAHs in the production process of enterprises (Liu et al., 2011). For example, Jungers et al. (2022) investigated the 16 priority-controlled PAHs levels of 24 pesticides, and found that the concentration of 11 PAHs exceeded 50 times of the standards, and three of them were more than the standards by over 1000 times, which resulted in the ratio of the PAHs to the threshold of toxicity was from 2.16 to 8288 times.

P2 Line 79-85:

“According to data from the National Bureau of Statistics of China (http://www.stats.gov.cn/), the total production of pesticides was 2.148 million tons in China in 2020. The number of pesticide manufacturing factories in China is approximately 1500~1700 (Wang et al., 2022). These results indicate that PAHs emissions from pesticide factories in China may be severe, and it is important to investigate the concentration and distribution characteristics of PAHs in typical pesticide factories.

Reference:

Jungers, G., Portet-Koltalo, F., Cosme, J., Seralini, G.E., 2022. Petroleum in Pesticides: A Need to Change Regulatory Toxicology. Toxics 10.

Liu, T., He, H., Sun, C., Wang, R., Cai, H., 2011. Distribution characteristics of PAHS in a brownfield in Changzhou. Environ. Chem. 30, 1456–1461 (in Chinese).

Wang, N., Shi, M., Wu, S., Guo, X., Zhang, X., Ni, N., Sha, S., Zhang, H., 2022. Study on Volatile Organic Compound (VOC) Emission Control and Reduction Potential in the Pesticide Industry in China. Atmosphere 13.

  1. Section 2.1. Study site and sample collection. How about the population density around the factory? It was suggested to briefly introduce the distribution of residents around the factory.

Response: Suggestion taken. The sentence has been revised as follows:

P3 Line 110-113:

“It is worth noting that there are a total of 10 villages within 7 km of the pesticide factory, of which the nearest is 1.5 km and the farthest is about 6.4 km away, and the total population of these villages is about 18000.”

  1. Line 147-148: “The concentrations of PAHs were stable throughout the sampling period (Fig. 2).” It’s not “concentration”. It may be the “composition proportion of PAH”.

Response: Suggestion taken. The sentence has been revised as follows:

P3 Line 177-178:

“The composition proportion of PAHs were stable throughout the sampling period (Fig. 2).

Reviewer 2 Report

This manuscript by Wang et al. assessed the concentration, composition, and health risk of 52 PM2.5-bound PAHs during the daytime and nighttime in the vicinity of a typical pesticide factory. I think the study is well designed and the manuscript is well written. I have several comments and recommend acceptance after a careful revision.

Line 37, the definition of PM2.5 is not correct. PM2.5 should be the particulate matter with aerodynamic diameters equal to or less than 2.5μm.

Section 2.1 it would be better if the authors supply a map to show the position of the factory, river, the location of the sampling site, and other buildings that described in the text.

Line 133, the authors used “significantly higher”, however, the term “significant” should be followed by the statistical analysis. If there was not statistical analysis, please remove the word. The word “significant” or “significantly” occurred in other places of the text, please check them.

The conclusion is too long, please shorten it and give a brief conclusion.

Author Response

We appreciate the editor and anonymous reviewer for their constructive comments and suggestions, which assist to improve the quality of our manuscript.

Reviewer #2:

This manuscript by Wang et al. assessed the concentration, composition, and health risk of 52 PM2.5-bound PAHs during the daytime and nighttime in the vicinity of a typical pesticide factory. I think the study is well designed and the manuscript is well written. I have several comments and recommend acceptance after a careful revision.

  1. Line 37, the definition of PM2.5 is not correct. PM2.5 should be the particulate matter with aerodynamic diameters equal to or less than 2.5μm.

Response: Suggestion taken. The sentence has been revised as:

P1 Line 37-39:

“Atmospheric particulate matter of diameter ≤2.5 μm (PM2.5) has become a global concern due to its adverse effects on the environment and human health (Shen et al., 2016; Sun et al., 2022).

  1. Section 2.1 it would be better if the authors supply a map to show the position of the factory, river, the location of the sampling site, and other buildings that described in the text.

Response: Suggestion taken. A map of the sampling area was provided in Fig. S1. The related description has been added below.

P3 Line 96-99:

“Atmospheric PM2.5 samples were obtained in the vicinity of a pesticide factory (34.37°N, 118.35°E) in Xuzhou, Jiangsu Province, China. The pesticide factory mainly produces insecticides, fungicides, herbicides, and chemical intermediates. The map near the sampling site was shown in Fig. S1.”

  1. Line 133, the authors used “significantly higher”, however, the term “significant” should be followed by the statistical analysis. If there was not statistical analysis, please remove the word. The word “significant” or “significantly” occurred in other places of the text, please check them.

Response: Suggestion taken. These results were compared to previous studies and could not be statistical analyzed. The word “significantly” was removed. Also, the word “significant” or “significantly” occurred in other places of the text were examined. The sentence has been revised as follows.

P4 Line 160-163:

“However, the concentrations are higher than those reported in studies assessing ambient air samples in Beijing (Σ61PAHs, 59.96 ng/m3; Lin et al., 2015), Xi’an (Σ52PAHs, 65.40 ng/m3; Sun et al., 2022), and Hong Kong (Σ46 PAHs, 4.67 ng/m3; Ma et al., 2016).”

  1. The conclusion is too long, please shorten it and give a brief conclusion.

Response: Suggestion taken. The brief conclusion has been revised as follows.

P11 Line 306-324:

“In the present study, daytime and nighttime PM2.5-bound PAHs emitted from a typical pesticide factory were detected. Their concentration, temporal variation, and health risk were assessed. The average concentration of 52 PAHs was 200.63 ± 159.42 ng/m3. Although Σ52PAHs varied during the sampling period, no significant differences were observed between daytime and nighttime samples. p-PAHs were the most abundant PAHs, accounting for 55.6% of the total PAH concentration. The proportion of high-molecular-weight (4–5-ring) PAHs was higher than that of low-molecular-weight PAHs. The analysis of diagnostic ratios suggested that the emissions from the pesticide factory are caused by production processes and coal combustion. The average ΣTEQ of the PAHs was 61.93 ± 46.46 ng/m3. These values are much higher than those observed in ambient air samples, indicating the potential risks of exposure to emissions from the pesticide factory. The ILCR of the PAHs was 2.33 × 10−3 for men and 2.53 × 10−3 for women. The LLE caused by the ΣPAHs was 11915 min (about 0.023 year) for men and 12952 min (about 0.025 year) for women. Among the four types of PAHs, the LLE caused by p-PAHs was the highest (11169 min for men and 12141 for women) and was 28 times and 33 times higher than the LLE caused by o-PAHs and n-PAHs, respectively. The LLE caused by a-PAHs was the lowest. The results of this study indicate the presence of elevated concentrations of PAHs in the emissions from the pesticide factory, which could potentially impact the health of individuals working in proximity to these emissions over the long term.”

Reviewer 3 Report

General comment

This paper reports a study of PAHs in PM2.5 collected near a pesticide factory in China discussing some aspects related to diagnostic ratios and to potential effects on health. The topic could be interesting in theory but after reading the paper I remained with several doubt regarding not convincing results and not focused approach.

The first point is that the sampling period is relatively limited and it is not clear is the samples were taken when the wind was blowing from the factory or not. No analysis of meteorological conditions is provided.

Measurements were taken for a limited period it could be sufficient to investigate the contribution of that specific source with a focused sampling strategy but this was not done or at least was not clear. However, it is well known that there are strong seasonality in the concentrations of PAHs and also in the ratios of congeners with different molecular weight so that the measurement period is not sufficient to furnish an overview of health risks.

The use of diagnostic ratios is rather qualitative and when it is said that a contribution from coal combustion and mixed sources is present it is not a very strong information. In addition, it is also mentioned that maybe the factory use coal for energy production. This should be a quite easy thing to check. In any case it is not an info related to pesticides.

For the reasons above, I suggest to reject the paper in the current form.

Author Response

We appreciate the editor and anonymous reviewer for their constructive comments and suggestions, which assist to improve the quality of our manuscript.

Reviewer #3:

This paper reports a study of PAHs in PM2.5 collected near a pesticide factory in China discussing some aspects related to diagnostic ratios and to potential effects on health. The topic could be interesting in theory but after reading the paper I remained with several doubt regarding not convincing results and not focused approach.

The first point is that the sampling period is relatively limited and it is not clear is the samples were taken when the wind was blowing from the factory or not. No analysis of meteorological conditions is provided.

Response: Suggestion taken. The weather conditions during the sampling period were supplemented in Table S1. The sampling site is located to the east of the pesticide factory, and based on the wind direction during the sampling period, most of PAHs can be considered to emit from the pesticide factory. The related statements have been supplemented as follows.

P3 Line 105-110:

“The weather conditions during the sampling period were shown in Table S1. During the sampling period, the weather was mainly sunny and the wind direction was mainly easterly. The sampling site was located at the east direction of the factory chimney. Therefore, the sampling site was downwind of the factory. Due to the impact of atmospheric transport, most of the PAHs in the collected samples were mainly come from the emission of the pesticide factory.”

Table S1. The meteorological parameters, including temperature (T), relative humidity (RH), wind speed (WS), and wind direction (WD) during the sampling period.

Date

T (℃)

RH (%)

WD

WS (m/s)

2023.3.15 day

5~14

70

Northeast

7

2023.3.15 night

6~14

75

East

5

2023.3.16 day

4~13

90

East

5

2023.3.16 night

4~13

83

East

5

2023.3.17 day

1~9

87

Northeast

4

2023.3.17 night

1~9

68

Northeast

1

2023.3.18 day

−1~15

70

Northwest

2

2023.3.18 night

1~15

70

Southwest

1

2023.3.19 day

2~17

70

Southeast

2

2023.3.19 night

2~17

62

Southeast

2

2023.3.20 day

3~18

66

Southeast

3

2023.3.20 night

3~18

76

Southeast

2

2023.3.21 day

7~18

84

East

3

2023.3.21 night

7~18

84

Northeast

3

2023.3.22 day

10~14

95

Northeast

3

2023.3.22 night

10~14

94

North

4

2023.3.23 day

8~17

45

Northeast

6

2023.3.23 night

8~17

35

East

4

Measurements were taken for a limited period it could be sufficient to investigate the contribution of that specific source with a focused sampling strategy but this was not done or at least was not clear. However, it is well known that there are strong seasonality in the concentrations of PAHs and also in the ratios of congeners with different molecular weight so that the measurement period is not sufficient to furnish an overview of health risks.

Response: Thanks for pointing out the issue. Even though the measurements in this study were taken for a limited period, the health risks assessment for PAHs should be reasonable. Here comes with the reasons.

First, considering the geographical location and surrounding conditions of the target pesticide factory, most of the PAHs in the sampling area should be emitted from the pesticide factory.

Second, according to the environment impact assessment statement of the pesticide factory, the production system of the factory is a continuous production method, with an annual operating time of 300 days and an annual working hours of 7200 hours. This indicates that PAHs emissions from the pesticide factory are stable. Therefore, the PAHs concentration in this region may not be strong seasonality.

The use of diagnostic ratios is rather qualitative and when it is said that a contribution from coal combustion and mixed sources is present it is not a very strong information. In addition, it is also mentioned that maybe the factory use coal for energy production. This should be a quite easy thing to check. In any case it is not an info related to pesticides.

Response: We agreed with the reviewer’s opinion. Although the diagnostic ratios of certain PAHs are widely employed to identify and distinguish the potential sources of PAHs, the shortcoming of this method is that it can only be used as a qualitative determination. In previous studies, positive matrix factorization (PMF) model has been used to quantify contribution from different sources (Ma et al., 2020; Sun et al., 2022). However, due to the limited number of samples, the PMF model could not be run, we used diagnostic ratios instead to determine the potential sources of PAHs in this study. In addition, since the sensitive constraints of the target factory, the sampling campaign can only be carried out at the boundary of the pesticide factory, rather than at production units. The total PAHs collected in this study should come from the boiler fuel combustion, solvent use, and production processes of the factory. In future research, we will attempt to collect samples in the boiler, production units, storage tanks, and confined spaces of the pesticide factory, which is of great significance to accurately evaluated the characteristics of PAHs emitted from different functional areas of the pesticide factory. To the best of our knowledge, there is limited research on PAHs from pesticide factory. This study provides a supplement to the concentration characteristics and health risks of PAHs emitted from pesticide factory.

Reference:

Ma, L., Li, B., Liu, Y., Sun, X., Fu, D., Sun, S., Thapa, S., Geng, J., Qi, H., Zhang, A., Tian, C., 2020. Characterization, sources and risk assessment of PM2.5-bound polycyclic aromatic hydrocarbons (PAHs) and nitrated PAHs (NPAHs) in Harbin, a cold city in Northern China. Journal of Cleaner Production 264.

Sun, J., Shen, Z., Zhang, T., Kong, S., Zhang, H., Zhang, Q., Niu, X., Huang, S., Xu, H., Ho, K.F., Cao, J., 2022. A com-prehensive evaluation of PM2.5-bound PAHs and their derivative in winter from six megacities in China: Insight the source-dependent health risk and secondary reactions. Environ Int 165, 107344.

For the reasons above, I suggest to reject the paper in the current form.

Response: The comments have been carefully addressed in the revised manuscript. We hope the revised version could meet the requirement of both reviewer and journal of toxics.

Reviewer 4 Report

It needs to emphasize the importance of air pollutants emitted from pesticide factories first in Introduction. Then how much PAH are released to the air with PM2.5 should be presented by references.

Since the health risks of PAHs have already been found in many studies, it would be better to present the particular points of PAHs releasing from pesticide factories. Chap. 3.4 is a just assessment of general PAHs, not from pesticide manufacturing process.

More details on sampling such as points, interval, amount, methods etc would be required.

Please provide a logical explanation for the comparison of night and day emissions. 

Minor editing of English language required.

Author Response

We appreciate the editor and anonymous reviewer for their constructive comments and suggestions, which assist to improve the quality of our manuscript.

Reviewer #4:

It needs to emphasize the importance of air pollutants emitted from pesticide factories first in Introduction. Then how much PAH are released to the air with PM2.5 should be presented by references.

Response: Suggestion taken. We have provided the impacts of the pesticide factories on the ambient air. However, to the best of our knowledge, research on PM2.5-bound PAHs emitted from pesticide factory is limited, the total PAHs emissions cannot be provided by references. The sentence has been revised as follows.

P2 Line 68-78:

“The pesticide factories contribute significantly to air pollutants emissions in the organic chemical industry (Wang et al., 2022), which is due to the fact that the various volatile and highly toxic substances, such as aromatic hydrocarbons, ketones and aldehydes are involved in the production of pesticides (Bari and Kindzierski, 2017; Masih et al., 2018). Some PAHs and coal tar containing PAHs are also important raw materials for pesticide production, which may lead to the emission of PAHs in the production process of enterprises (Liu et al., 2011). For example, Jungers et al. (2022) investigated the 16 priority-controlled PAHs levels of 24 pesticides, and found that the concentration of 11 PAHs exceeded 50 times of the standards, and three of them were more than the standards by over 1000 times, which resulted in the ratio of the PAHs to the threshold of toxicity was from 2.16 to 8288 times.”

Reference:

Bari, M.A., Kindzierski, W.B., 2017. Concentrations, sources and human health risk of inhalation exposure to air toxics in Edmonton, Canada. Chemosphere 173, 160-171.

Jungers, G., Portet-Koltalo, F., Cosme, J., Seralini, G.E., 2022. Petroleum in Pesticides: A Need to Change Regulatory Toxicology. Toxics 10.

Liu, T., He, H., Sun, C., Wang, R., Cai, H., 2011. Distribution characteristics of PAHS in a brownfield in Changzhou. Environ. Chem. 30, 1456–1461 (in Chinese).

Masih, A., Lall, A.S., Taneja, A., Singhvi, R., 2018. Exposure levels and health risk assessment of ambient BTX at urban and rural environments of a terai region of northern India. Environ Pollut 242, 1678-1683.

Wang, N., Shi, M., Wu, S., Guo, X., Zhang, X., Ni, N., Sha, S., Zhang, H., 2022. Study on Volatile Organic Compound (VOC) Emission Control and Reduction Potential in the Pesticide Industry in China. Atmosphere 13.

Since the health risks of PAHs have already been found in many studies, it would be better to present the particular points of PAHs releasing from pesticide factories. Chap. 3.4 is a just assessment of general PAHs, not from pesticide manufacturing process.

Response: Thanks for pointing out the issue. However, since the sensitive constraints of the target factory, the sampling campaign can only be conducted at the boundary of the pesticide factory, rather than at production units. This study only assessed the health risks of total PAHs, which released into the air due to boiler fuel combustion, solvent use, and production processes in the factory. To the best of our knowledge, there is limited research on PAHs from pesticide factory. Therefore, the current study focused on the impact of emissions from the whole pesticide factory rather than the different production links. In future research, we will attempt to collect samples in the boiler, production units, storage tanks, and confined spaces of the pesticide factory, which is of great significance to accurately evaluated the characteristics of PAHs emitted from different functional areas of the pesticide factory.

More details on sampling such as points, interval, amount, methods etc would be required.

Response: Suggestion taken. More information about the sampling were supplemented.

P3 Line 114-123:

“At the sampling site near the pesticide factory, a medium-volume PM2.5 sampler with a flow rate of 50 L/min was used to collect PM2.5 samples on 47-mm prebaked quartz-fiber filters (Whatman, Maidstone, Kent, UK). Samples were collected continuously daily during the daytime (8 a.m. to 8 p.m.) and nighttime (8 p.m. to 8 a.m.) between March 15 and 23, 2023. Finally, a total of nine daytime samples and eight nighttime samples were collected for further analysis. A field blank sample was collected at the same sampling site. All sampled filters were transported at −18 ℃, and stored in a freezer at the same temperature in the laboratory. The quartz-fiber filters were used in accordance with the quality assurance and control protocols described in detail by Cao et al. (2012) and Shen et al. (2017).”

Please provide a logical explanation for the comparison of night and day emissions. 

Response: Suggestion taken. As described in section 2.1, considering the geographical location and surrounding conditions of the target pesticide factory, most of the PAHs in the sampling area were emitted from the pesticide factory. Therefore, the diurnal variation concentration of PAHs in this study should be influenced by the production process of the pesticide factory. According to the engineering and construction technical data, and environment impact assessment statement of the target pesticide factory, the production system of the factory is a continuous production method, with an annual operating time of 300 days and an annual working hours of 7200 hours. Therefore, the factory conducts production activities both day and night, which may be the reason for the lack of significant differences in PAHs concentration between day and night. We have provided explanation for the comparison of night and day emissions as follows.

P4 Line 170-174:

“Although Σ52PAHs varied substantially during the sampling period, no significant differences were observed between daytime and nighttime concentrations, which may be related to the production process of the pesticide factory. The basic factory information collected included the environment impact assessment statement, and engineering data, which show that the production system of the factory is continuous operation.”

Comments on the Quality of English Language: Minor editing of English language required.

Response: Suggestion taken. The English language of manuscript has been professionally edited.

Round 2

Reviewer 3 Report

Authors somewhat improved the paper during revision and provided answers to my questions. I still believe that the limited number of samples gives strong limitation to the usability of results. However, it can be accepted inthe current form.